# Modelling protein complexes with crosslinking mass spectrometry and deep learning

Kolja Stahl [1], Robert Warneke[2], Lorenz Demann[2], Rica Bremenkamp[2], Björn Hormes[2], Oliver Brock [3,4], Jörg Stülke [2] ✉ & Juri Rappsilber [1,5,6] ✉

Scarcity of structural and evolutionary information on protein complexes poses a challenge to deep learning-based structure modelling. We integrate experimental distance restraints obtained by crosslinking mass spectrometry (MS) into AlphaFold-Multimer, by extending AlphaLink to protein complexes. Integrating crosslinking MS data substantially improves modelling performance on challenging targets, by helping to identify interfaces, focusing sampling, and improving model selection. This extends to single crosslinks from whole-cell crosslinking MS, opening the possibility of whole-cell structural investigations driven by experimental data. We demonstrate this by revealing the molecular basis of iron homoeostasis in *Bacillus subtilis*.

Solving the structure of protein complexes is key to understanding life at the molecular level. The advent of deep learning-based methods has significantly improved the reliability of single protein structure prediction[1]. However, the general lack of co-evolutionary information and the lower number of solved structures makes predicting the structure of protein complexes a more difficult task[2]. Experimental distance restraints, such as proximal residue pairs revealed by crosslinking mass spectrometry (MS), can identify interactions[3–5] and their interfaces[6]. The experimental information can supplement information derived from evolutionary relationships in guiding structure prediction[7].

In our previous work, we improved the modelling of challenging single protein targets by leveraging distance restraints in AlphaLink[7], a deep learning-based method that integrates crosslinking MS data directly into the pair representation of AlphaFold2[1] to bias the prediction. Here, we extend AlphaLink to protein complexes. Protein complexes pose a much harder problem because on top of predicting the structure of the individual chains, predicting interactions requires searching a space that grows exponentially with the size of the complex. Crosslinking MS data can help to cut down the search space and reduce the dependency of the prediction on evolutionary information.

In this study, we train with succinimidyl 4,4-azipentanoate (SDA) crosslinks but also successfully test with a different crosslinker. SDA is a soluble crosslinker that provides restraints between residues receptive to its first reaction step, Lys, Ser, Thr, Tyr, and other amino acids on protein surfaces[8] and is readily available. SDA crosslinks are of lower resolution (expected < 25 Å Cα-Cα) compared to the photo-AA crosslinks (expected < 15 Å Cα-Cα) we used on single proteins. The lower resolution poses an additional challenge and the weaker evolutionary signal changes the balance of the data from evolutionary information towards crosslink-based distance restraints, which requires more heavy lifting of the network.

We evaluate AlphaLink on challenging heteromeric CASP15 (Critical Assessment of Structure Prediction[9]) and antibody-antigen targets with simulated crosslinks. Moreover, we validate AlphaLink on in-cell crosslinking MS data[5] from *Bacillus subtilis* that used a different crosslinker, and a virally modified Cullin4-RING ubiquitin ligase (CRL4) complex[10,11]. We focus the evaluation on heteromeric assemblies since homomers pose a different challenge. The inherent ambiguity of self-links in homo-multimeric assemblies rarely permits distinguishing inter- from intra-chain restraints[6].

[1]Technische Universität Berlin, Chair of Bioanalytics, Berlin, Germany. [2]Georg-August-Universität Göttingen, Department of General Microbiology, Institute for Microbiology & Genetics, GZMB, Göttingen, Germany. [3]Technische Universität Berlin, Robotics and Biology Laboratory, Berlin, Germany. [4]Science of Intelligence, Research Cluster of Excellence, Berlin, Germany. [5]Si-M/"Der Simulierte Mensch", a Science Framework of Technische Universität Berlin and Charité - Universitätsmedizin Berlin, Berlin, Germany. [6]Wellcome Centre for Cell Biology, University of Edinburgh, Edinburgh, UK. ✉e-mail: jstuelk@gwdg.de; juri.rappsilber@tu-berlin.de

## Results and discussion

### Integrating crosslinking MS data substantially improves the prediction quality of challenging CASP15 targets

AlphaLink with distance restraints vastly outperforms AlphaFold-Multimer[2] on challenging heteromeric CASP15 targets. It achieves similar or better results than the best performing methods in CASP15, which used up to 120x more sampling to derive predictions. Integrating simulated SDA crosslinks in the modelling of eight challenging heteromeric CASP15 targets (H1129, H1134, H1140, H1141, H1142, H1144, H1166, H1167) substantially improved the DockQ[12] score from 0.14 to 0.62 on average, compared to the AlphaFold-Multimer baseline (Fig. 1a) which matches the average DockQ = 0.62 of the best predictions in CASP15. For comparison, AFsample[13], one of the top-performing methods in CASP15, averages a DockQ score of 0.56. Except for H1142, including crosslinking MS data always produced at least acceptable solutions according to DockQ score (DockQ ≥ 0.23[12]). To ensure comparability, we used the same input features as AlphaFold-Multimer (multiple sequence alignments (MSAs) and templates) and fine-tuned AlphaLink on the v2 network weights of AlphaFold-Multimer. Compared to the baseline AlphaFold-Multimer prediction, we increased recycling iterations from 3 to 20 and the number of samples from 25 to 200 (except for H1129 due to computing limitations) to be more in line with methods in CASP15. Increasing sampling and recycling iterations improve the predictions for targets where the crosslinks allow a high degree of flexibility (e.g., H1141) and side chain placement. As a control, Supplementary Fig. 1 shows AlphaLink outperforming AlphaFold-Multimer with the same recycling iterations = 3 and a comparable number of samples (10 samples for AlphaLink vs 25 for AlphaFold-Multimer).

AlphaLink underperformed compared to the top groups in CASP15 for three specific targets (H1129, H1134, H1141). In contrast, AFsample excelled on these targets and ranked among the best performers. AFsample increases random exploration in baseline AlphaFold-Multimer by sampling more (up to ~24000 samples for H1166) and turning on dropout during inference which randomly removes information to create more diversity. This strategy can be implemented in AlphaLink. Leveraging crosslink data provides two key benefits: it reduces the search space by concentrating sampling on regions of interest, thereby requiring fewer samples, and it supplements evolutionary information with additional, complementary data. This is particularly valuable for challenging targets, such as antigen-antibody pairs.

For H1129, Yang et al. achieved a DockQ over 0.6 by expanding the MSA search and including the monomer predictions as a template. We also noticed that AlphaFold-Multimer predicts chain B of H1129 separately better than within the complex (0.89 vs 0.79), suggesting a potential problem with the fold-and-dock approach of AlphaFold-Multimer and thus AlphaLink which is partially side-stepped by including the monomer prediction as a template. Including the monomer prediction of chain B of H1129 as a template drastically improved the prediction quality (Supplementary Fig. 2).

H1134 has only a few contacts in the interface and is more flexible (Supplementary Fig. 3a), plausibly explaining the large spread in the DockQ scores in CASP15 (Supplementary Fig. 3b). Such flexible targets especially benefit from increased sampling and recycling. In the case of H1134, the model confidence is also less discriminative (Supplementary Fig. 4). Crosslinks can improve model selection in this case by selecting first by crosslink satisfaction and second by model confidence (see the accordingly modified AlphaLink* in Supplementary Fig. 3b). Training AlphaLink from scratch in the future could improve this by better reflecting crosslinking information in the model confidence.

Originally, in the case of H1141, SDA crosslinks were not restrictive enough (Supplementary Fig. 1). We observe two clusters that differ in the relative orientation of the subunits. Both clusters satisfy all crosslinks (blue in Supplementary Fig. 5). However, the better cluster

(model confidence 0.86 versus 0.38) corresponds to the crystal structure (DockQ score 0.72). Increasing sampling and recycling helped to select the correct cluster.

The test set comprised both dimers and multimeric assemblies. The simulated crosslink-derived distance restraints had 10% sequence coverage and 20% false-discovery rate[14], resulting in 33 links in the median per protein-protein interaction (including 7 false links, in the median). For comparison purposes, we selected the best predictions based on the highest model confidence[2] (0.8 * interface predicted TM-score (ipTM) + 0.2 * predicted TM-score), which allows us to pick close to the best model (Supplementary Fig. 6). Indeed, ipTM and DockQ correlate (ipTM 0.6 equates roughly to DockQ 0.4)[2]. We compare the performance of AlphaLink with and without crosslinks to exclude the influence of other parameters, such as having trained AlphaLink on larger crops than AlphaFold-Multimer v2.2 and the additional fine-tuning. Indeed, we see that the observed improvements are the result of integrating crosslinks (Supplementary Fig. 7).

### Crosslinking MS data improve modelling of challenging antibody-antigen targets

We extended the evaluation by predicting in addition to CASP15 targets also the interactions of 32 recent antibody-antigen targets from the Structural Antibody Database (SAbDab)[15] (Fig. 1b). Antibody-antigen targets are challenging because the co-evolutionary signal is lower[16]. Here, AlphaLink improves the DockQ on average from 0.29 to 0.59 compared to AlphaFold-Multimer. To limit compute, we used three recycling iterations and five samples for both methods. These results confirm what we have seen for CASP15 where the largest improvements were observed on nanobody-antigen (yellow shaded area in Fig. 1a) and antibody-antigen targets (red shaded area in Fig. 1a)[16]. In these cases, crosslinking MS data drastically aided prediction. AlphaFold-Multimer models are incorrect for 5 out of 6 targets (DockQ < 0.23[12]) while AlphaLink generates at least medium quality models for 5 out of 6 targets (DockQ > 0.49[12]). Notably, for H1142, H1166, and H1167, AlphaLink produced better median score predictions than the top-ranked CASP15 submissions (Figs. 1a, c, d). The DockQ score for H1142 improved from 0.01 for AlphaFold-Multimer and 0.1 in CASP15 to 0.855 by leveraging crosslinking MS data. All true links are satisfied, while 100% of the false links are rejected. This demonstrates a high resilience of AlphaLink towards noise when predicting protein complexes, as was already observed for single proteins[7]. Similarly, the DockQ score for H1166 improved from 0.22 (AlphaFold-Multimer) to 0.8 (AlphaLink), again with crosslink satisfaction and noise rejection being 100%.

### A single crosslink obtained in cells can dramatically improve model quality

Encouraged by the success of AlphaLink with simulated data, we modelled 135 dimeric protein-protein interactions (PPIs) with real data[5] from *Bacillus subtilis* cells crosslinked in situ with disuccinimidyl sulfoxide (DSSO)[17]. This soluble crosslinker provides restraints between primarily Lys but also Ser, Thr, and Tyr groups. The data are sparse (median 1 crosslink per PPI), however even one crosslink can drastically improve the results (Fig. 2a). For example, the model confidence of the CodY-YppF interaction[5,18] improves from 0.25 to 0.81 based on a single crosslink (Fig. 2b). For RpoA-RpoC (PDB 6WVK), the DockQ improves from 0.003 to 0.69 based on four in-cell crosslinks (Fig. 2c). Each crosslink would have sufficed to improve model quality substantially (DockQ 0.69-0.7). We note that changed fine-tuning (larger crops) in AlphaLink2 and AlphaFold-Multimer v2.3 alone leads to some improvements, returning correct models in 5 out of 10 cases for RpoA-RpoC, while adding crosslinks improved this to 10 out of 10.

AlphaLink is fine-tuned on model_1, in comparison, AlphaFold-Multimer predicts the targets with 5 different networks. Using different networks improves the model confidence on average by 0.03 points

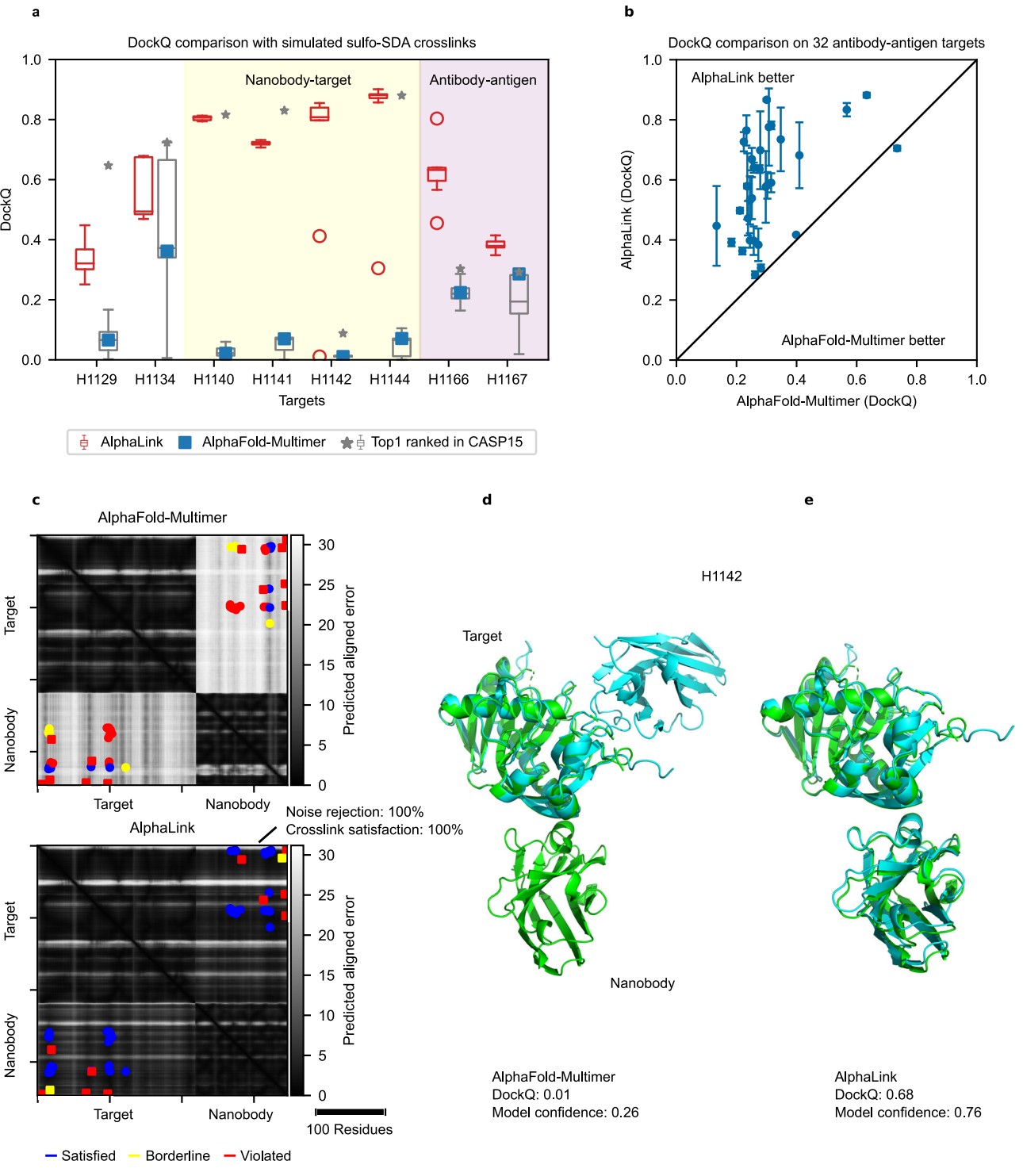

**Fig. 1 | AlphaLink performance on simulated data. a** DockQ comparison on 8 heteromeric CASP15 targets with simulated SDA crosslinks. AlphaLink boxplot shows the highest confidence model out of 200 predictions for N = 10 randomly sampled crosslink sets. The top1 ranked predictions (grey) correspond to submissions that were ranked first by the participants, best prediction is highlighted with an asterisk, AlphaFold-Multimer in blue. The yellow shaded area highlights nanobody, the red shaded area antibody targets. For boxplots, the line shows the median and the whiskers represent the 1.5x interquartile range. **b** DockQ comparison of AlphaLink and AlphaFold-Multimer on 32 antibody-antigen targets (SAb-Dab). Error bars represent the 95% confidence interval over N = 10 randomly sampled crosslink sets. Points show the mean. **c** PAE map comparison of AlphaFold-Multimer and AlphaLink for H1142. **d** AlphaFold-Multimer prediction for H1142 (cyan) aligned to the crystal structure (green). **e** AlphaLink prediction for H1142 (cyan) aligned to the crystal structure (green).

(Supplementary Fig. 8). As seen with the CASP15 targets, crosslinks help to focus sampling on the interesting regions, reducing the amount of sampling required. Overall, integrating crosslinking MS data increases the median model confidence from 0.42 (AlphaFold-Multimer) to 0.6

(AlphaLink). With AlphaLink, 12 additional interactions (total of 46, gain of 35%) reach model confidence > 0.75. Remarkably, the AlphaLink network was not trained on DSSO data, which differ in linked residues, linker length, and density of data from simulated SDA data. Thus, our

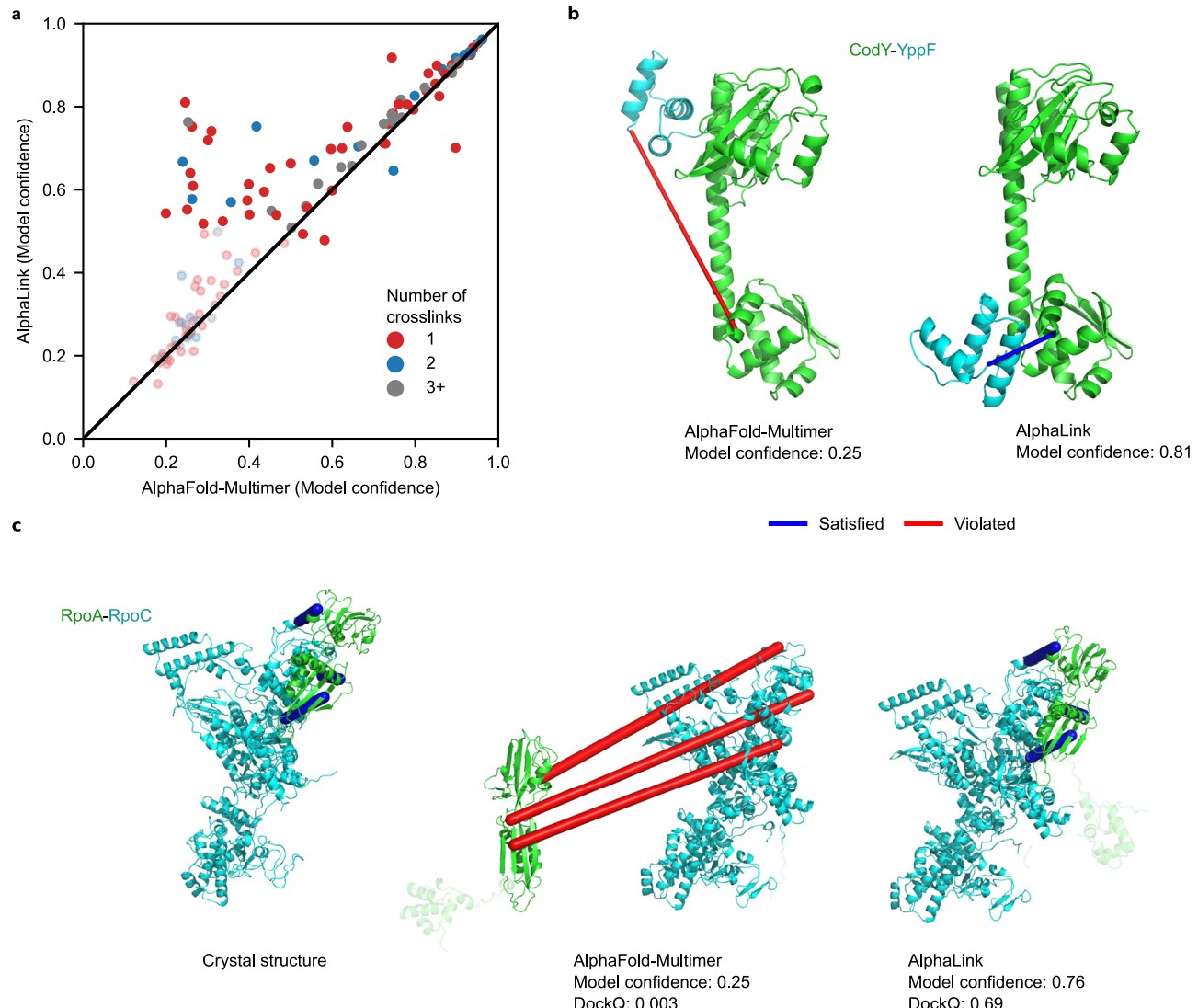

**Fig. 2 | AlphaLink performance on real data from *Bacillus subtilis*. a** Highest model confidence of AlphaLink v2.2 (N = 5) vs AlphaFold-Multimer v2.2 (M = 5) on the *Bacillus subtilis* data on K = 135 dimeric protein-protein interactions. Low confidence predictions for both methods (model confidence < 0.5) have higher transparency. **b** AlphaFold-Multimer (left) and AlphaLink (right) prediction for CodY-YppF with annotated crosslinks (blue - satisfied, red - violated). **c** Comparison of crystal structure (left), AlphaFold-Multimer prediction (middle) and AlphaLink prediction (right) for RpoA-RpoC with annotated crosslinks (blue - satisfied, red - violated). We omitted one crosslink that is not covered by the crystal structure.

results suggest the broad applicability of AlphaLink to varying cross-linker chemistry and the typically low data densities achieved in large-scale and whole-cell crosslinking MS experiments.

## AlphaLink reveals the long-enigmatic molecular basis of iron homoeostasis in *B. subtilis*

Our recent in-cell crosslinking investigation of protein interactions in *B. subtilis*[5] revealed that the conserved bacterial global regulator for iron homoeostasis, Fur, interacts with an essential protein of unknown function, YlaN[18,19]. We confirm here this interaction in a bacterial two-hybrid assay (Fig. 3a). In addition, and in agreement with published structures, both Fur[20] and YlaN[21] exhibited self-interaction. Analyses of a Fur-regulated promoter (see Fig. 3b) and in vitro binding of Fur to the promoter region (Fig. 3c) demonstrate that YlaN acts as an antagonist of Fur that prevents it from binding to its DNA targets and thus allows expression of the Fur-repressed genes in the absence of iron (see also Supplementary Discussion for detail). Accordingly, we renamed YlaN to Fpa (Fur protein antagonist). This is corroborated by the observation, that the Fpa knockout is viable if iron(III) is added to the medium[22]. We resorted next to AlphaLink to elucidate the structural

basis of Fpa inhibiting Fur. The AlphaLink prediction (model confidence: 0.75) of the *B. subtilis* Fur-dimer (Fig. 3d) resembles a V-shaped conformation, similar to other Fur proteins which interact with DNA[20]. AlphaLink predicts a single conformation (model confidence 0.84) consistent with the crosslinking MS data between Lys-74 in the DNA-binding domain of Fur and the Lys residues 23 and 26 of Fpa[5,23] (Fig. 3d). This benefited from the integration of crosslinks into the prediction, as AlphaFold-Multimer predicts two conformations for the Fur-Fpa complex, one of which is not supporting an interaction (model confidence 0.41). Fpa directly engages the DNA-binding domains of Fur and disassembles the Fur dimerisation interface, resulting in a breaking of the functional dimer and a complete re-orientation of the DNA-binding domain (see http://www.subtiwiki.uni-goettingen.de/v4/predictedComplex?id=168 and http://www.subtiwiki.uni-goettingen.de/v4/predictedComplex?id=169 for an interactive display of the Fur dimer and the Fur-Fpa complex as well as Supplementary Movie 1). This structure clearly shows why Fur bound by Fpa is incapable of binding DNA. To verify our AlphaLink model of the Fur-Fpa complex, we tested the interaction of a mutant Fur (Fur*, K18A R23E Y60E) version with Fpa. These surface mutations induce a charge reversal and

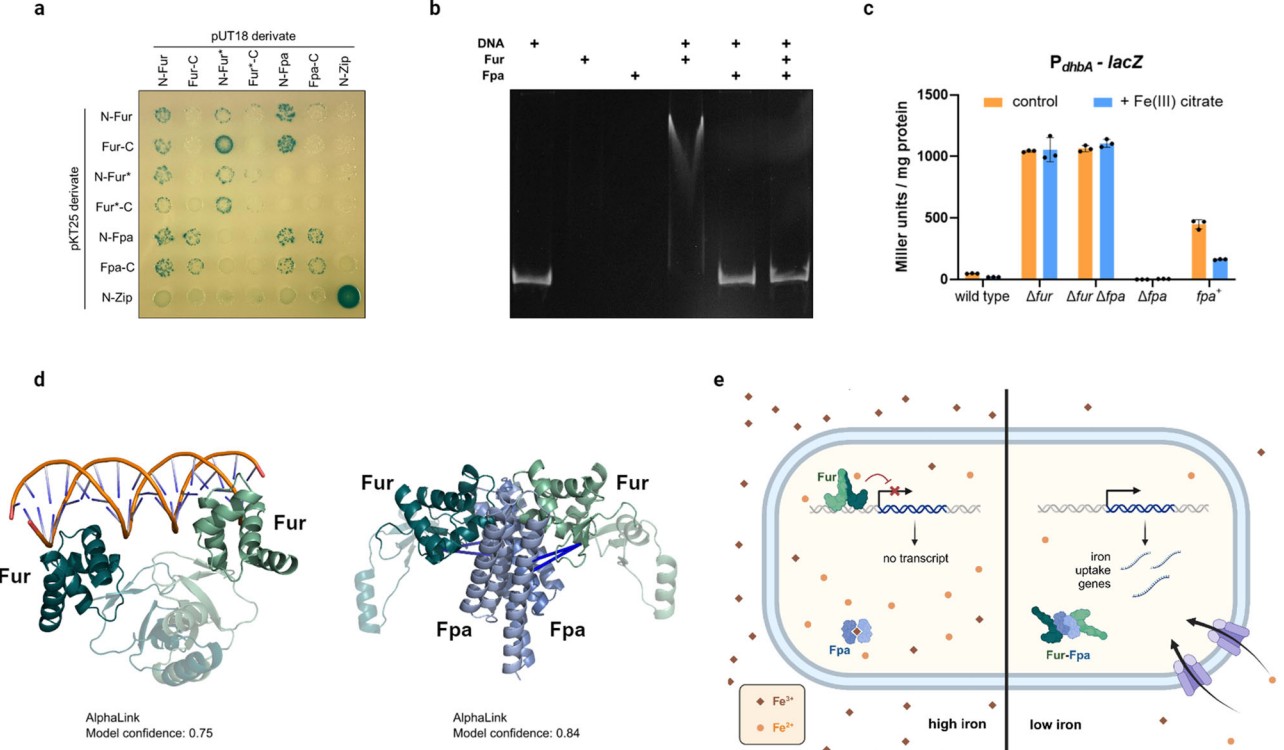

**Fig. 3 | The Fpa-Fur interaction controls iron homoeostasis in *B. subtilis*.**
**a** Bacterial two-hybrid assay to test the interaction between Fur and Fpa. N- and C-terminal fusions of both proteins as well as the mutant Fur* protein to the T18 or T25 domain of the adenylate cyclase (CyaA) were created and the proteins were tested for interaction in *E. coli* BTH101. Blue colonies indicate an interaction that results in adenylate cyclase activity and subsequent expression of the reporter β-galactosidase. **b** Gel electrophoretic mobility shift assay of Fur binding to *dhbA* promoter fragments in the absence of iron. The components added to the assays are shown above the gels. The same results were obtained in three independent experiments. **c** The impact of Fpa on *dhbA* promoter activity. Strains carrying a *dhbA-lacZ* fusion were cultivated in LB with or without added ferric citrate, and promoter activities were determined by quantification of β-galactosidase activities.

The values are averages of three independent biological replicates. Standard deviations are shown. **d** AlphaLink predicts the structure of the Fur-Fpa complex. Left: AlphaLink prediction (model confidence: 0.75) of the Fur dimer. Right: AlphaLink prediction (model confidence: 0.84) of the Fpa-Fur complex with crosslinking MS data. The Fur dimer undergoes a large conformational change. The dimerization region is shown in transparent. All crosslinks (shown as blue lines) are satisfied in the prediction. **e** Model for the control of Fur activity by Fpa. At high iron concentrations, the Fur dimer binds its target DNA sequences in the promoter regions of genes involved in iron homoeostasis. The Fpa protein binds ferrous iron and is unable to interact with Fur. If iron gets limiting, apo-Fpa forms a complex with Fur, resulting in the release of Fur from its DNA targets, and thus in expression of genes for iron uptake.

destroy the salt bridges at different contact points of the binding interface between Fur and Fpa. The Fur* protein showed self-interaction as well as an interaction with the wild type Fur protein. In contrast, no interaction was observed between the mutant Fur* protein and Fpa (see Fig. 3a), confirming the predicted AlphaLink model. The proposed rotation and accompanying re-orientation of the DNA-binding domains of Fur explain the loss of the DNA-binding activity of Fur upon interaction with Fpa. Interestingly, the nature of the inducer molecule that causes the release of Fur from DNA in the absence of iron has long been debated. Although direct binding of ferrous iron as a co-repressor has been suggested[24], there is no experimental evidence for such a mechanism. On the contrary, Fur alone binds a promotor in the absence of iron in the buffer (Fig. 3b). We show here that it requires the recruitment of the Fur DNA-binding domains by the iron(II)-sensitive Fpa[25] to secure the release of Fur from DNA and thereby iron homoeostasis in low iron conditions (Fig. 3e).

**Crosslinking MS-driven modelling of a multi-protein complex**
Finally, we challenged AlphaLink to model, with the help of real sulfo-SDA data, the 6-subunit CRL4^DCAF1-CtD/Vpr_mus/SAMHD1 assembly[11] (360 kDa, 3118 AA) (Fig. 4), using a crystal structure and cryo-EM density[11] as ground truth. The accessory protein Vpr from certain simian immunodeficiency viruses targets the DCAF1 substrate receptor of host Cullin4-RING ubiquitin ligases (CRL4), to recruit and mark the restriction factor SAMHD1 for proteasomal degradation and thus

to stimulate virus replication[11,26]. The v2.2 network weights of AlphaFold-Multimer fail to return meaningful structures of this assembly. This is overcome by moving to v2.3 network weights, which have been trained on larger complexes. Leveraging crosslinks improves the model confidence from 0.56 (AlphaFold-Multimer v2.3) to 0.64 (AlphaLink v2.3) (Fig. 4a). The CRL4 subunit CUL4A shows substantial movements, resulting in three conformational states according to cryo-EM data (Fig. 4b). AlphaFold-Multimer predicts a contact between CUL4A and DCAF1 that is not supported by the cryo-EM density (Fig. 4c). By contrast, AlphaLink predicts no contact (Fig. 4d). The density doesn't contain SAMHD1 (shown in light red in Figs. 4c, d). SAMHD1 has both crosslinks to CUL4A (grey) and DDB1 (purple) which influences the arrangement. In addition, crosslinks allow AlphaLink to position the viral Vpr protein inside the experimental density in agreement with the crystal structure (PDB 6ZX9) (Figs. 4e, green, f), while AlphaFold-Multimer places Vpr incorrectly. Our predicted Vpr-DCAF1 interaction model achieved a DockQ score of 0.56 compared to 0.04 by AlphaFold-Multimer (Fig. 4f).

## Discussion
Our findings demonstrate the successful extension of AlphaLink, an experiment-assisted AI approach, to predict protein complex structures. We can now start visualising at pseudo-atomic resolution protein-protein interactions inside cells, by leveraging the very scarce data of whole-cell crosslinking MS. With this breakthrough,

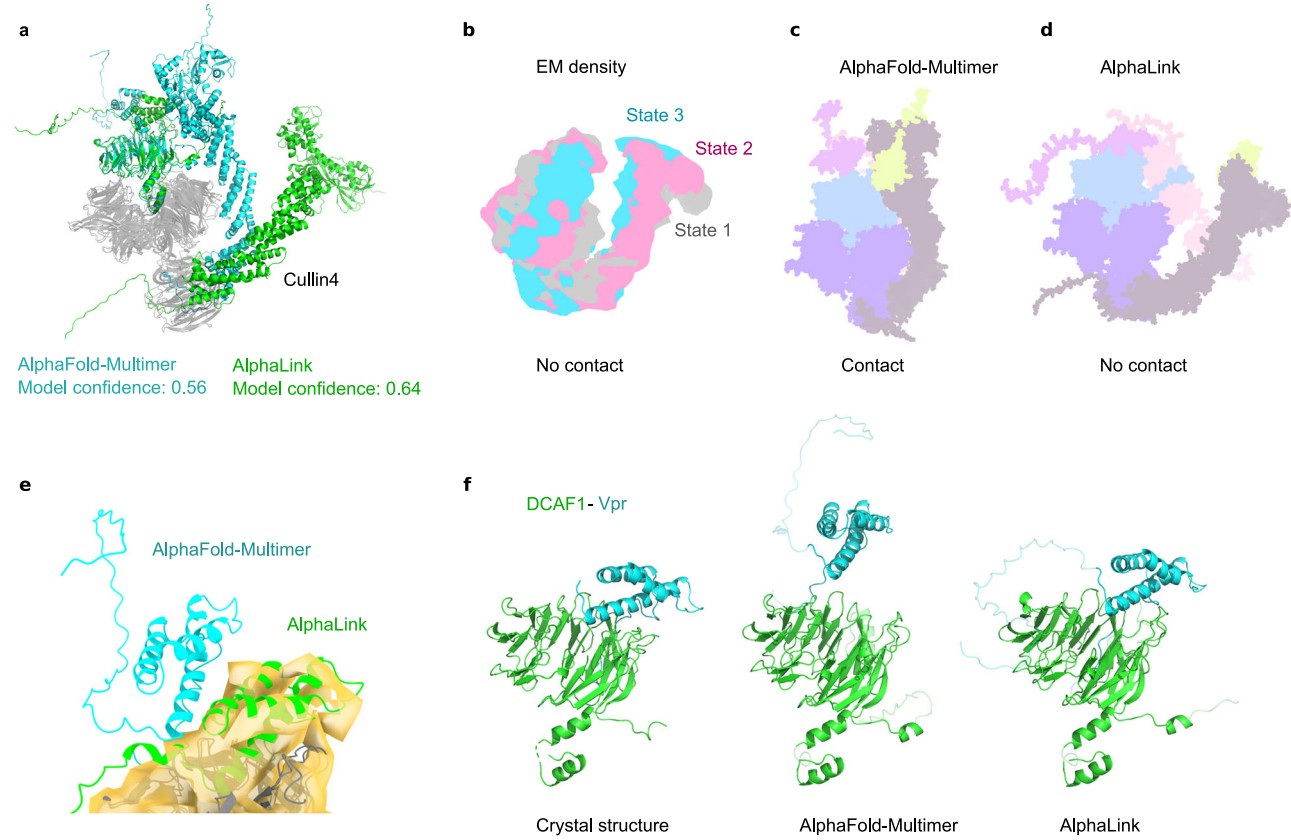

**Fig. 4 | AlphaLink performance on real data from CRL4 with v2.3 weights.**
**a** AlphaFold-Multimer (v2.3) (cyan) and AlphaLink (v2.3) (green) prediction for a virally modified CRL4 assembly. SAMHD1 is not visualised for clarity. **b** Schematic view of the three states of Cullin4 observed in the EM density (EMD-10612 to EMD-10614). SAMHD1 is not part of the density. **c** Schematic view of the AlphaFold-Multimer prediction, showing a contact not present in the density (**b**). **d** Schematic view of the AlphaLink prediction. SAMHD1 is shown in light red. **e** Placement of the viral Vpr protein for AlphaFold-Multimer (cyan) and AlphaLink (green) in the EM density. **f** Predictions of DCAF1-Vpr compared to the crystal structure (PDB 6ZX9).

protein-protein interaction studies move from delivering links (who) to structures (how), inside cells and at scale. Importantly, in-cell crosslinking reveals protein interactions that are often lost upon cell lysis[5]. While lysis is frequently required for large-scale study of protein complexes, doing so without prior crosslinking challenges, especially transient and fragile features of complexes. AlphaLink with whole-cell crosslinking will therefore accelerate discovering hitherto hidden aspects of biology, to advance our understanding of life and widen our therapeutic options in cases of disease.

## Methods

### Crosslink simulation
We simulate SDA crosslinks with XWalk[27] with a 25 Å Cα-Cα cutoff and trypsin digestion. XWalk identifies cross-linkable residue pairs based on solvent accessibility, crosslinker, and peptide digestion. We add FDR = 20% of noise to match the expected FDR. At least one crosslink is always incorrect, the actual FDR can therefore be much higher. False positive crosslinks can be any residue pair > 25 Å Cα-Cα, where one residue is a Lys, Ser, Thr, and Tyr. In real data, there is often an affinity towards Lys and clustering of crosslinks. These biases are not reflected in the simulation since we uniformly subsample the crosslink candidates. This doesn't play a big role in training, and we can see that it translates to real data, but it may give better coverage during testing than what we can normally expect. The link-level FDR is simulated by shuffling the crosslinks and counting the number of incorrect links observed so far. The coverage is set to 10% and corresponds to the sequence coverage based on the longer sequence. We sample inter- and intra-protein crosslinks independently.

### Integration of crosslinks
We integrate crosslinks in AlphaLink the same way we did in the monomer version[7]. We add a crosslink embedding layer to the neural network that projects the soft label contact map into the 128-d z-space. The projection is added to the pair representation (z). This way, the crosslinking information influences retrieval of the co-evolutionary information and the coupled updates with the MSA representation enables noise rejection.

### Fine-tuning of alphafold-multimer
We switched to Uni-Fold[28] since OpenFold[29] didn't support multimers. To avoid training Uni-Fold from scratch, we fine-tune the weights provided by Deepmind. For v2 we use AlphaFold-Multimer 2.2.4 weights as the starting point (https://github.com/deepmind/alphafold/releases/tag/v2.2.4). AlphaFold-Multimer was trained on PDBs deposited before 2018-04-30, predating CASP13. For v3, we use the AlphaFold-Multimer 2.3.0 weights (https://github.com/deepmind/alphafold/releases/tag/v2.3.0). AlphaFold-Multimer was trained on PDBs deposited before 2021-09-30, predating CASP15. The networks are refined on 11424 protein complexes with a total of 34054 chains from the DIPS-Plus[30] training set with simulated SDA crosslinking data. DIPS-Plus contains PDBs deposited before May 2021, predating CASP15. We use Uni-Fold v2.1.0 (https://github.com/dptech-corp/Uni-Fold/releases/tag/v2.1.0). MSAs were generated with the reduced database setting. We train and test with model_1. Since we focus on heteromers, we sample homomeric crosslinks like heteromeric crosslinks during training to have more samples.

For training, we follow the refinement training regime outlined in the AlphaFold-Multimer paper but expand the crop size to 640AA to increase interface exposure and the number of crosslinks we see during training. We train on 4 A100 GPUs for 10 days. We used early stopping on the validation set which consists of proteins from CAMEO[31] released after 2022.

## Evaluation set up

For the comparison, we use the official predictions from CASP15. The CASP15 targets were classified as TBM/FM (template-based modelling / free modelling), meaning that there is a partial template for a subunit and/or the assembly. Our main comparison point is NBIS-AF2-multimer which corresponds to standard AlphaFold-Multimer v2. We use the same MSAs as NBIS-AF2-multimer provided by Arne Elofsson (http://duffman.it.liu.se/casp15/). We increased the recycling iterations to 20, the original comparison can be found in Supplementary Fig. 1. We randomly sample 10 crosslink sets with 10% coverage and 20% FDR for each target and predict each with a different seed for a total of 200 seeds (10 in the original comparison). We only relax the best sample (chosen by model confidence) per crosslink set. The maximum MSA cluster size is 512 sequences.

For the SAbDab comparison, we compile a new data set consisting of 33 recent antibody-antigen targets between 01-01-2022 and 11-10-2023 which represent challenging protein complexes due to the lower evolutionary signal. We only include targets which have a crystal structure with a resolution of 3 Å or better and a single assembly, to simplify the evaluation. We do not remove targets that may have homologues in the training set.

The primary evaluation metric is the DockQ score. We compute the DockQ scores for AlphaLink with the official CASP15 evaluation scripts (https://git.scicore.unibas.ch/schwede/casp15_ema).

The DockQ score is the average of the fraction of native contacts ($F_{nat}$), the interface RMSD (iRMS)[32], and the ligand RMSD (LRMS)[32]. The interface includes all contacting residues. Residues are in contact if they are from different chains and at least one heavy atom is within 5 Å. The iRMS increases the cutoff to 10 Å. The $F_{nat}$ then corresponds to the recall of the native contacts. The iRMS is the RMSD of the backbone atoms in the interface. The LRMS is the RMSD after superimposing the larger of the two structures onto the smaller one. The final DockQ score is the average DockQ score over all interfaces for protein complexes with more than two chains.

We only relax the best prediction per crosslink subset to save compute time. Relaxing only slightly changes the final scores.

For the *Bacillus subtilis* predictions, we use the same MSAs as O'Reilly et al[5]. We predict the targets again with AlphaFold-Multimer v2.2 to be comparable. The results with the original AlphaFold-Multimer v2.1 predictions from the study are shown in Supplementary Fig. 9. The model confidence might not be comparable. Except for a few targets, there are no crystal structures available for *Bacillus subtilis* which is why we have to resort to model confidence as an indicator of improvement. Supplementary Fig. 10a shows the correlation between the model confidence and the DockQ score and further the relationship between the model confidences of AlphaLink and AlphaFold-Multimer with respect to the DockQ score (Supplementary Fig. 10b). Although, on these hard targets, the model confidence is an overestimation, a better model confidence generally translates into a better DockQ score.

For the Cullin4 complex, we use the v3 weights to predict the structures. We always predict the full complex and use the same MSAs for both AlphaFold-Multimer and AlphaLink. We compare the prediction of DCAF1-Vpr to the crystal structure (PDB 6ZX9) with the T4 tag removed. The EM densities correspond to the EMDB accession codes: EMD-10611 (core), EMD-10612 (conformational state-1), EMD-10613 (state-2) and EMD-10614 (state-3). There are on average 21 crosslinks per interface.

The structures are visualised with PyMol v2.5.0 and ChimeraX 1.7.1.

## Strains, media and growth conditions

*E. coli* DH5α and Rosetta DE3 (28) were used for cloning and for the expression of recombinant proteins, respectively. All *B. subtilis* strains used in this study are derivatives of the laboratory strain 168. They are listed in Supplementary Table 1. *B. subtilis* and *E. coli* were grown in Luria-Bertani (LB) or in sporulation (SP) medium[33,34]. For growth assays and the in vivo interaction experiments, *B. subtilis* was cultivated in LB, SP, or CSE-Glc minimal medium[34,35]. CSE-Glc is a chemically defined medium that contains sodium succinate (6 g/l), potassium glutamate (8 g/l), and glucose (1 g/l) as the carbon sources[35]. Iron sources were added as indicated. The media were supplemented with ampicillin (100 μg/ml), kanamycin (50 μg/ml), chloramphenicol (5 μg/ml), or erythromycin and lincomycin (2 and 25 μg/ml, respectively) if required. LB and SP plates were prepared by the addition of Bacto Agar (Difco) (17 g/l) to the medium. All oligonucleotides used in this study are listed in Supplementary Table 2.

## DNA manipulation

Transformation of *E. coli* and plasmid DNA extraction were performed using standard procedures[33]. All commercially available plasmids, restriction enzymes, T4 DNA ligase and DNA polymerases were used as recommended by the manufacturers. *B. subtilis* was transformed with plasmids, genomic DNA or PCR products according to the two-step protocol[34]. Transformants were selected on SP plates containing erythromycin (2 μg/ml) plus lincomycin (25 μg/ml), chloramphenicol (5 μg/ml), kanamycin (10 μg/ml), or spectinomycin (250 μg/ml). DNA fragments were purified using the QIAquick PCR Purification Kit (Qiagen, Hilden, Germany). DNA sequences were determined by the dideoxy chain termination method[33].

## Construction of mutant strains by allelic replacement

Deletion of the *fur* and *fpa* genes was achieved by transformation of *B. subtilis* 168 or GP879 with a PCR product constructed using oligonucleotides to amplify DNA fragments flanking the target genes and an appropriate intervening resistance cassette[36]. The integrity of the regions flanking the integrated resistance cassette was verified by sequencing PCR products of about 1100 bp amplified from chromosomal DNA of the resulting mutant strains.

## Phenotypic analysis

In *B. subtilis*, amylase activity was detected after growth on plates containing nutrient broth (7.5 g/l), 17 g Bacto agar/l (Difco) and 5 g hydrolysed starch/l (Connaught). Starch degradation was detected by sublimating iodine onto the plates.

Quantitative studies of *lacZ* expression in *B. subtilis* were performed as follows: cells were grown in CSE-Glc or LB medium supplemented with iron sources as indicated. Cells were harvested at OD600 of 0.5 to 0.8. b-Galactosidase specific activities were determined with cell extracts obtained by lysozyme treatment[34]. One unit of β-galactosidase is defined as the amount of enzyme which produces 1 nmol of o-nitrophenol per min at 28 °C.

## Plasmid constructions

To express the Fur and Fpa proteins carrying a N-terminal His-tag in *E. coli*, the *fur* and *fpa* genes were amplified using chromosomal DNA of *B. subtilis* 168 as the template and appropriate oligonucleotides that attached specific restriction sites to the fragment. Those were: BamHI and XhoI for cloning *fur* in pET-SUMO (Invitrogen, Germany), and BamHI and SalI for cloning *fpa* in pWH844[37]. The resulting plasmids were pGP3589 and pGP2583 for Fur and Fpa, respectively.

For overexpression of *fpa* in *B. subtilis*, we constructed plasmid pGP3897. For this purpose, the *fpa* gene was amplified and cloned between the BamHI and SalI site of the expression vector pBQ200[38].

Plasmid pAC7[39] was used to construct a translational fusion of the *dhbA* promoter region to the promoterless *lacZ* gene. For this purpose, the promoter region was amplified using oligonucleotides that attached EcoRI and BamHI restriction to the ends of the products. The fragments were cloned between the EcoRI and BamHI sites of pAC7. The resulting plasmid was pGP3594.

## Protein expression and purification

*E. coli* Rosetta(DE3) was transformed with the plasmid pGP371[40], pGP2583, and pGP3589 encoding His-tagged versions of PtsH, Fpa, and Fur, respectively. For overexpression, cells were grown in 2x LB and expression was induced by the addition of isopropyl 1-thio-β-D-galactopyranoside (final concentration, 1 mM) to exponentially growing cultures ($OD_{600}$ of 0.8). The His-tagged proteins were purified in 1x ZAP buffer (50 mM Tris-HCl, 200 mM NaCl, pH 7.5). Cells were lysed by four passes (18,000 p.s.i.) through an HTU DIGI-F press (G. Heinemann, Germany). After lysis, the crude extract was centrifuged at $46,400 \times g$ for 60 min and then passed over a $Ni^{2+}$nitrilotriacetic acid column (IBA, Göttingen, Germany). The proteins were eluted with an imidazole gradient. After elution, the fractions were tested for the desired protein using SDS-PAGE. The purified proteins were concentrated in a Vivaspin turbo 15 (Sartorius) centrifugal filter device (cut-off 5 or 50 kDa). The protein samples were stored at −80 °C until further use. The protein concentration was determined according to the method of Bradford[41] using the Bio-Rad dye binding assay and bovine serum albumin as the standard.

## Electromobility shift assay (EMSA) with DNA

To analyse the binding of Fur to the *dhbA* promoter region, we performed EMSA assays with a 284 bp *dhbA* promoter fragment that carries the Fur binding site and purified Fur, Fpa, and PtsH proteins. 200 ng of DNA and 80 pmol of the proteins were used. The samples were first prepared without the proteins only with DNA, buffer and water and heated for 2 minutes at 95 °C. Then the proteins were added in different combinations and the samples were incubated for 30 minutes at 37 °C. Meanwhile, the EMSA gels were applied to a pre run at 90 V for 30 minutes immersed in TBE buffer (28). Afterwards, 2 μl of the loading dye were added and the samples were loaded into the gel pockets. The gel was run for 3 hours at 110 V. Then, the gels were immersed in TBE containing HDGreen® fluoreszence dye (Intas, Germany). After 2 minutes the gels were photographed under UV light.

## Bacterial two-hybrid assay

Primary protein-protein interactions were identified by bacterial two-hybrid (BACTH) analysis[42]. The BACTH system is based on the interaction-mediated reconstruction of *Bordetella pertussis* adenylate cyclase (CyaA) activity in *E. coli* BTH101. Functional complementation between two fragments (T18 and T25) of CyaA as a consequence of the interaction between bait and prey molecules results in the synthesis of cAMP, which is monitored by measuring the β-galactosidase activity of the cAMP-CAP-dependent promoter of the *E. coli lac* operon. Plasmids pUT18C and p25N allow the expression of proteins fused to the T18 and T25 fragments of CyaA, respectively. For these experiments, we used the plasmids pGP3868-pGP3875, which encode N-and C-terminal fusions of T18 or T25 to *fur* and *fpa*. The plasmids were obtained by cloning the *fur* and *fpa* between the KpnI and BamHI sites of pUT18C and p25N[42]. The mutant *fur** allele was purchased from Eurofins Genomics (Germany) and then amplified and cloned as the wild type *fur* gene. The resulting plasmids were then used for co-transformation of *E. coli* BTH101 and the protein-protein interactions were then analysed by plating the cells on LB plates containing 100 μg/ml ampicillin, 50 μg/ml kanamycin, 40 μg/ml X-Gal (5-bromo-4-chloro-3-indolyl-ß-D-galactopyranoside), and 0.5 mM IPTG (isopropyl-ß-D-thiogalactopyranoside). The plates were incubated for a maximum of 36 h at 28 °C.

## Reporting summary

Further information on research design is available in the Nature Portfolio Reporting Summary linked to this article.

## Data availability

The AlphaLink models generated in this study based on experimental crosslinks have been deposited as integrative/hybrid models in PDB-Dev[43] under accession codes PDBDEV_00000221 and PDBDEV_G_1000003. The previously published data we used in this study are available in the PRIDE and EM database under accession code PXD020453 (Cullin4 crosslinks), EMD-10611 (core), EMD-10612 (state-1), EMD-10613 (state-2) and EMD-10614 (state-3); JPST001796 and PXD035508 JPST001797 and PXD035519 JPST001791 and PXD035362 (*Bacillus subtilis*). Source data are provided in this paper.

## Code availability

The code for AlphaLink is deposited at https://github.com/Rappsilber-Laboratory/AlphaLink2. Model weights were deposited at Zenodo: https://doi.org/10.5281/zenodo.8007238.

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

## Acknowledgements

We thank Andrea Graziadei and David Schwefel for their feedback on the manuscript and David Schwefel for helping to identify potential mutations. Christina Herzberg is acknowledged for the help with some reporter assays. We are grateful to the Uni-Fold team for providing a fully opensource and trainable reimplementation of AlphaFold-Multimer (https://github.com/dptech-corp/Uni-Fold). Figure 3/panel e, created with BioRender.com, released under a Creative Commons Attribution-NonCommercial-NoDerivs 4.0 International license. This research was supported by the Wellcome Trust [Grant number 227166] (JR, KS) and the Deutsche Forschungsgemeinschaft (DFG, German Research Foundation) under Germany´s Excellence Strategy – EXC 2008 – project number 390540038 (JR) – UniSysCat (JR) and under Germany's Excellence Strategy – EXC 2002/1 "Science of Intelligence" – project number 390523135 (OB), as well as CRC1565 [Grant number 469281184] (JS) and SPP1879 [project number STU214/ 16-2] (JS). The Wellcome Centre for Cell Biology is supported by core funding from the Wellcome Trust [203149].

## Author contributions

Software development: K.S. Data Analysis: K.S., L.D., R.B., R.W., B.H. Supervision: J.R., O.B., J.S. Supplement information: L.D., R.B., R.W., B.H., J.S. Writing- initial draft: K.S., J.R., O.B. Writing- editing and revision: K.S., J.R., O.B., L.D., R.B., R.W., B.H., J.S.

## Funding

## Competing interests

The other authors declare no competing interests.
