## [Transparent Peer Review file · Nature Communications]

Modelling protein complexes with crosslinking mass spectrometry and deep learning

Corresponding Author: Professor Juri Rappsilber

Figures originally included in the author's rebuttal have been redacted from this file.

Version 0:

Reviewer comments:

Reviewer #1

(Remarks to the Author)

The study by Stahl, Brock, and Rappsilber describes the inclusion of distance constraints from cross-linking mass spectrometry in the machine learning-based structure prediction tool AlphaFold Multimer. The authors have previously implemented this for single-protein structure prediction by AlphaFold2. With the present additions, the previous tool, AlphaLink, additionally considers crosslinks between protein pairs when choosing the highest-scoring structure prediction for multiprotein complexes. The authors demonstrate improved structure prediction using several multiprotein complexes with experimentally determined structures, using DockQ scores as read-out. They then apply the extended AlphaLink to a previously published whole-cell crosslinking dataset and demonstrate that even single-crosslinks can improve the accuracy of the predicted structures.

The study is clearly described and (at least according to what I can judge from my expertise) conforms to the highest standards of technical accuracy. I fully believe the authors' claims, the issue with the work is rather that it feels very much like an extension of previous work (even if it required changing the base and training the network), and the most significant finding, that is, how distance restraints can improve AF predictions, has already been reported (Stahl et al, Nature Biotech 2023). In this sense, the study would be more suitable for a specialist journal.

To me, the real novelty of the study is its application to whole-cell crosslinking data, which, as the authors note themselves in the abstract, opens the possibility of whole-cell structural investigations that are biased by experimental data. However, the corresponding section in the manuscript is extremely short, and basically only states that even a single crosslink improves model accuracy. Is there a way to extract more biologically relevant information from the improved models, particularly when compared to the group's previous work in this area (O'Reilly et al, Ref 5)? If the authors can show that the improved models translate to novel biological insights, that would greatly elevate the general interest.

Reviewer #2

(Remarks to the Author)

General comments.

The method presented (AlphaLink) is an interesting extension of the authors' previous work on single chain complexes. However, too little data is used for evaluation and the results compared to other computational methods are not satisfactory as AlphaLink is outperformed in most cases (4 vs 3 cases where AlphaLink is better). Another similar method called ColabDock also exists (<https://www.biorxiv.org/content/10.1101/2023.07.04.547599v1>), but was not used in the evaluation.

It is important to consider data overlaps between all different methods that have been integrated into AlphaLink. This is not assessed. The methods section is very short and is lacking important information about fine-tuning (e.g. loss curves?) the data sets used, their overlaps and compositions. In general, a description of exactly what information goes into AlphaLink and how it is provided is lacking.

Significant work is needed to assess the issues with evaluation and explaining exactly what advancements AlphaLink provides compared to other available methods.

Specific comments.

1. The legend in Figure 1a mentions predictions from AlphaFold, but what is meant here is really AlphaFold-multimer(?) which is not the same method. This is an issue throughout the manuscript which makes it difficult to follow.
2. Seven hard targets from CASP15 are evaluated. Four of these are nanobody interactions and two are antibody interactions. Compared to the best performance in CASP15, AlphaLink only outperforms predictions from AlphaFold-multimer in three cases. In four cases (H1129, H1134, H1140, H1141) does AFsample (the best method in CASP15) outperform AlphaLink.
3. The star indicating the top ranked model from CASP15 is missing for H1167 so this may really only be true for two targets (H1142 and H1166). The failure to outperform available computational methods raises questions for the utility of AlphaLink.
4. I assume that the reason that AlphaLink only outperforms purely computational methods in the antibody-antigen cases is because there is very little (and different) coevolution there. AlphaLink may still be useful in these cases, but the data presented is too little to properly assess this.
5. AlphaLink uses simulated crosslinks from XWalk, but no mention of what data this method is trained on and the overlap to the evaluation set is presented.
6. AlphaLink is fine-tuned starting from AFM on a dataset with simulated cross-links, but no information on the composition of this or the overlap to the CASP15 targets or the training set of AFM is provided.
7. To prove the utility of AlphaLink, I suggest extending the analysis significantly. Extract a dataset of antibody-antigen interactions which have not been seen by AFM during training and compare the performance. It is not possible to assess the performance on so few targets, especially since there is much more data available (1000s of structures in SAbDab: <https://opig.stats.ox.ac.uk/webapps/sabdab-sabpred/sabdab>). Other methods for antibody-antigen structure prediction exist as well and these should also be included in the evaluation (e.g. <https://www.nature.com/articles/s41467-023-38063-x>).
8. Figure 1b. This plot is confusing (the axis labels are the same). What are you really plotting here? I suggest plotting the model confidence vs the DockQ after selecting for the highest model confidence and after running a single prediction. This is shown for H1134 in Extended data Figure 2 where one can see that the model confidence correlates poorly with the DockQ (although it is hard to tell since all 100 predictions are plotted on top of each other).
9. Figure 2a. Again, the same axis-labels. I am not sure what this plot is supposed to display, but being "better" is arbitrary. It would be more informative to plot the model confidence vs the number of cross links and analyse the correlation between the two. This together with model confidence vs DockQ answers the questions of if the confidence measure can be trusted and if the cross-links really help.

In addition, plot the model confidence vs the DockQ and colour by the number of cross links.

Reviewer #3

(Remarks to the Author)

The authors have submitted a natural extension of Alphaslink to the modeling of heteromeric complexes. They demonstrate that high quality models can be generated in a fraction of the time of a "normal" AlphaFold Multimer modeling event. This claim is established with a very nice simulation/sampling of crosslinking data but also backed up in select cases with real (sparse!) data collected from other studies.

In general I find this a very useful and exciting addition, and a very realistic one. Computational methods will likely (always) benefit from the inclusion of real structural data and crosslinking MS is a natural source for such information.

I have only a few matters to raise:

1. As I say, the simulations are a nice way to illustrate the benefits of Alphaslink, but some additional information is needed. If I understand correctly, the authors use a span of ~25 angstroms and random multiple samplings of 10 crosslinks, from a set that samples 10% of sequence (at a 20% FDR). I like the challenge presented by these numbers. However, does the sampling of the crosslinks reflect the bias that we often see in real data? First at the level of residue (K much more preferred than STY) and second at the level of structure (links clustered in one area). Additional discussion around the limitations of the simulation would be valuable.
2. The first paragraph after the introduction strikes a rather discordant note. Any claim of "vastly outperforming" should be made after the data are presented, to avoid biasing the reader.
3. And I contest the very next statement that claims also that the results are similar or better on CASP15 targets than the best performing algorithms that participated in CASP15. Figure 1a shows a mixed bag, with 50% of the CASP15 algorithms outperforming Alphaslink. Please adjust the language on this.
4. And more importantly on this topic, what is to be learned about how and why these algorithms outperform Alphaslink? I notice in Extended Data Figure 6 that strongly increasing the sampling/recycling helps only a bit (on one example at least).

The authors are perfectly positioned to comment on benefits/weaknesses of approaches.

5. Figures:

- a. 2C. Is there an actual interaction between RpoA and RpoC in the AlphaFold model? The interface is obscured by the crosslinks. Can this be made more visible?
- b. 3D. I'm a bit puzzled by this success. Can the authors comment how the crosslinking preserved an open state when AlphaFold (which AlphaLink is based on) drives a closed form. Near as I can tell, the orientation of the lower hinge looks the same for both AlphaFold and AlphaLink. Can you prevent an interaction if crosslinks occupy an interface? Would be nice to see some commentary on this and a supporting figure with the crosslinks mapped.
- c. Extended Data figure 1. Better axis labels would be nice.
- d. Extended Data figure 6. What does "more" mean exactly? Better labels here too, please.

Author Rebuttal letter:

Reviewer #1 (Remarks to the Author):

The study by Stahl, Brock, and Rappsilber describes the inclusion of distance constraints from cross-linking mass spectrometry in the machine learning-based structure prediction tool AlphaFold Multimer. The authors have previously implemented this for single-protein structure prediction by AlphaFold2. With the present additions, the previous tool, AlphaLink, additionally considers crosslinks between protein pairs when choosing the highest-scoring structure prediction for multiprotein complexes. The authors demonstrate improved structure prediction using several multiprotein complexes with experimentally determined structures, using DockQ scores as read-out. They then apply the extended AlphaLink to a previously published whole-cell crosslinking dataset and demonstrate that even single-crosslinks can improve the accuracy of the predicted structures.

The study is clearly described and (at least according to what I can judge from my expertise) conforms to the highest standards of technical accuracy. I fully believe the authors' claims, the issue with the work is rather that it feels very much like an extension of previous work (even if it required changing the base and training the network), and the most significant finding, that is, how distance restraints can improve AF predictions, has already been reported (Stahl et al, Nature Biotech 2023). In this sense, the study would be more suitable for a specialist journal.

This work builds on previous work rather than extending it, as it wasn't obvious that the monomer approach would translate. Protein complex prediction is much more difficult because in addition to predicting the individual chains, predicting interactions makes it necessary to search a space that grows exponentially with the size of the complexes. Crosslinks are obviously super helpful here because they can cut down the search space. But the nature of the data is very different. Evolutionary data is generally sparse in the interface and the SDA crosslinks used in this study are lower in resolution (< 25Å CA-CA) compared to the photo-AA crosslinks used in the previous work (< 15Å CA-CA) which are much closer to co-evolutionary data, the main source of information. Because of this, the neural network has to do more heavy lifting here. We added text to point out these challenges.

To me, the real novelty of the study is its application to whole-cell crosslinking data, which, as the authors note themselves in the abstract, opens the possibility of whole-cell structural investigations that are biased by experimental data. However, the corresponding section in the manuscript is extremely short, and basically only states that even a single crosslink improves model accuracy. Is there a way to extract more biologically relevant information from the improved models, particularly when compared to the group's previous work in this area (O'Reilly et al, Ref 5)? If the authors can show that the improved models translate to novel biological insights, that would greatly elevate the general interest.

We respectfully disagree. This work constitutes a surprising and substantial advancement over the current technical state of the field. We would like to raise the reviewer's attention to the fact that a single crosslink from in-cell data can substantially improve model accuracy. Whether in such a proof-of-concept study also novel biology was produced or not is of no relevance to the technical advancement. We feel not alone in this view as Nature Communications published many papers reporting on computational progress without new biology, e.g., <https://www.nature.com/articles/s41467-022-33729-4> and <https://www.nature.com/articles/s41467-023-38063-x>. Also, Nature Methods did so: <https://www.nature.com/articles/s41592-024-02174-0>. Asking us to add biology comes as a surprising decision in this context, also because it is based on the comment of a reviewer that is neither skilled to judge the challenges that we had to overcome in our technical solution nor the significance of our finding for the structure prediction field.

Anyway, we extended the manuscript by adding work on YlaN-Fur to experimentally confirm our structure. The structural model predicted by AlphaLink, which agrees with the crosslinking data, shows how the YlaN-Fur interaction regulates iron homeostasis. YlaN engages the DNA interaction domain of Fur and splits the Fur homodimer, rendering Fur incompetent to bind DNA, repressing the genes necessary for iron uptake. We introduce point mutations in Fur based on our YlaN-Fur complex structure to successfully destroy the predicted interaction and validate our structural and mechanistic model of YlaN-mediated regulation of iron homeostasis in *B. subtilis*.

Reviewer #2 (Remarks to the Author):

General comments.

The method presented (AlphaLink) is an interesting extension of the authors' previous work on single chain complexes. However, too little data is used for evaluation and the results compared to other computational methods are not satisfactory as AlphaLink is outperformed in most cases (4 vs 3 cases where AlphaLink is better). Another similar method called ColabDock also exists (<https://www.biorxiv.org/content/10.1101/2023.07.04.547599v1>), but was not used in the evaluation.

We deliberately kept the parameters (and input data) in Fig. 1a in line with the baseline method AlphaFold-Multimer to show that including crosslinking MS data improves the prediction over the baseline. The other methods in CASP15, e.g., AFSample used as much as 2400x the number of samples in comparison to AlphaLink, in addition to other changes. We slightly increased the parameters now for AlphaLink to show a fairer comparison. The original comparison is moved to the supplement. The results further improved and show much more parity on the cases where AlphaLink was outperformed. Note that the approach taken by AFSample will not scale well for larger complexes since sampling will become prohibitively expensive, e.g., the prediction of a 6000 AA complex takes around 30 hours for a single sample. Being able to reduce the number of samples significantly by focusing the sampling on the interesting regions becomes more and more important with increasing complex sizes. That being said, the strategy taken by AFSample is compatible with AlphaLink and could be adopted. Also note that no single method in CASP15 outperformed all other methods.

We extended the evaluation to 33 challenging antibody-antigen complexes, here AlphaLink also outperforms AlphaFold-Multimer by a large margin.

We have compared our approach with all methods participating in CASP15. ColabDock appeared one month after us on BioRxiv and constitutes work that followed our work.

It is important to consider data overlaps between all different methods that have been integrated into AlphaLink. This is not assessed. The methods section is very short and is lacking important information about fine-tuning (e.g. loss curves?) the data sets used, their overlaps and compositions. In general, a description of exactly what information goes into AlphaLink and how it is provided is lacking.

Indeed, it is important to avoid potential biases and overfitting. We did early stopping on the validation set. We extended the data set description in methods to clarify the cutoff dates and exclude/ indicate potential overlaps. For the fine-tuning, we already show the results of AlphaLink with and without crosslinking MS data to make sure that the observed performance improvements are not due to additional training and using bigger crops (in comparison to AlphaFold-Multimer v2, Extended Data Figure 7). There is no data overlap between the AlphaLink and AlphaFold-Multimer training set and CASP15 targets, the new SAbDab targets, or Cullin4; since the training sets predate the data. Except for a couple of targets, there are no crystal structures for *Bacillus subtilis* complexes, therefore there is only a potential overlap with self-distillation. The targets that have crystal structures might have been in the training set of AlphaFold-Multimer but this is negligible because it would affect both AlphaFold-Multimer and AlphaLink in the same way.

Significant work is needed to assess the issues with evaluation and explaining exactly what advancements AlphaLink provides compared to other available methods.

Specific comments.

1. The legend in Figure 1a mentions predictions from AlphaFold, but what is meant here is really AlphaFold-multimer(?) which is not the same method. This is an issue throughout the manuscript which makes it difficult to follow.

Sorry for the confusion! We had defined in the manuscript that AlphaFold refers to AlphaFold-multimer. To avoid confusion, we explicitly call it now AlphaFold-Multimer or AF-Multimer in short.

2. Seven hard targets from CASP15 are evaluated. Four of these are nanobody interactions and two are antibody interactions. Compared to the best performance in CASP15, AlphaLink only outperforms predictions from AlphaFold-multimer in three cases. In four cases (H1129, H1134, H1140, H1141) does AFsample (the best method in CASP15) outperform AlphaLink. We didn't make sufficiently clear that this experiment is a control to show the improvement over baseline AlphaFold-multimer on the same input data and with the same setting. Even though AFsample uses baseline AlphaFold-multimer (+ dropout at inference time) the settings are vastly different from the standard settings in the baseline.

We can control the comparison with AlphaFold-multimer since the MSAs and templates are available. We show that AlphaLink outperforms AlphaFold-Multimer on all eight hard CASP15 targets (in the median) with the same input features and number of recycling iterations, even though AlphaFold-Multimer uses slightly more samples (25 vs 10) and more models (5 vs 1).

We cannot control for AFsample since the input data is not available and we lack the compute power. AFsample diverged greatly from the standard settings. To be more comparable, we increased the number of samples from 10 to 200 and the recycling iterations from 3 to 20 which is still much lower than the up to 24000 samples for AFsample. We moved the control experiment to the supplement.

Note that we could adopt the AFsample setting and sampling strategy in AlphaLink because AFsample makes only minor changes to standard AlphaFold-Multimer (dropout during inference) and is therefore compatible with AlphaLink. But this approach doesn't scale well. It quickly becomes infeasible to generate large amounts of samples. For example, Cullin4 takes 16 h to predict. Integrating crosslinks on the other hand, allows us to focus on the interesting regions and to drastically reduce the amount of samples required (see H1166 where AFsample failed with 24000 samples).

3. The star indicating the top ranked model from CASP15 is missing for H1167 so this may really only be true for two targets (H1142 and H1166).

The was unfortunately concealed by the AlphaFold-multimer box. We fixed it.

The failure to outperform available computational methods raises questions for the utility of AlphaLink.

In our new data AlphaLink is at the same level or better than the best submission to CASP15. No alternative approach achieved this, as the best prediction is not from the same lab.

4. I assume that the reason that AlphaLink only outperforms purely computational methods in the antibody-antigen cases is because there is very little (and different) coevolution there. AlphaLink may still be useful in these cases, but the data presented is too little to properly assess this.

This is indeed the underlying premise of AlphaLink. Combining experimental data (crosslinking MS) and co-evolutionary information will improve challenging targets, esp., those with little co-evolutionary data. We added a new dataset (Fig. 1b) with more challenging targets to highlight the utility of AlphaLink.

5. AlphaLink uses simulated crosslinks from XWalk, but no mention of what data this method is trained on and the overlap to the evaluation set is presented.

There is no overlap because XWalk is not trained on anything. It simulates crosslinks based on a given PDB.

6. AlphaLink is fine-tuned starting from AFM on a dataset with simulated cross-links, but no information on the composition of this or the overlap to the CASP15 targets or the training set of AFM is provided.

There is no overlap, both training sets (AlphaLink and AlphaFold-Multimer) are from

2021, predating the benchmark data. We added the cutoff dates to the methods section.

7. To prove the utility of AlphaLink, I suggest extending the analysis significantly. Extract a dataset of antibody-antigen interactions which have not been seen by AFM during training and compare the performance. It is not possible to assess the performance on so few targets, especially since there is much more data available (1000s of structures in SAbDab: <https://opig.stats.ox.ac.uk/webapps/sabdab-sabpred/sabdab>).

Unfortunately, most structures in the SAbDab were likely part of the training set of AlphaFold-Multimer and would constitute overlap. We compiled a new data set with 33 antibody-antigen targets from the last two years, which were released after the training set of both AlphaFold-Multimer and AlphaLink. To simplify evaluation, we only used targets which have a single biological assembly.

Other methods for antibody-antigen structure prediction exist as well and these should also be included in the evaluation (e.g. <https://www.nature.com/articles/s41467-023-38063-x>).

The suggested methods are specialised methods that solve a different problem, still e.g., in the case of IgFold, it was outperformed by AlphaFold-Multimer. We (and AlphaFold-Multimer) solve the general problem of predicting interactions. The antibody-antigen targets were interesting to us because they are challenging due to the limited evolutionary information.

8. Figure 1b. This plot is confusing (the axis labels are the same). What are you really plotting here? I suggest plotting the model confidence vs the DockQ after selecting for the highest model confidence and after running a single prediction. This is shown for H1134 in Extended data Figure 2 where one can see that the model confidence correlates poorly with the DockQ (although it is hard to tell since all 100 predictions are plotted on top of each other). Figure 1b compared the DockQ scores of two AlphaLink settings with one another, see labels inside the plot. We removed this plot and replaced it with new data on targets from SAbDab.

It was shown in the AlphaFold-Multimer paper that the ipTM (> 0.6) correlates well with the DockQ score which in turn is close to the model confidence. Extended Data Fig. 6 shows the correlation between the pick with the highest model confidence and the pick with the best DockQ score.

Extended Data Fig. 4 shows an example where the correlation is poor but crosslink satisfaction can help to identify good predictions.

9. Figure 2a. Again, the same axis-labels. I am not sure what this plot is supposed to display, but being "better" is arbitrary. It would be more informative to plot the model confidence vs the number of cross links and analyse the correlation between the two. This together with model confidence vs DockQ answers the questions of if the confidence measure can be trusted and if the cross-links really help.

In addition, plot the model confidence vs the DockQ and colour by the number of cross links.

Figure 2a compares the model confidence of AlphaFold (x-axis) and AlphaLink (y-axis). The regions are labelled accordingly inside the plot. To make it clearer, we now include the method names directly in the labels. We removed "better" from the labelling.

Unfortunately, for *Bacillus subtilis* there are no crystal structures for protein complexes except for few cases, which is why we cannot report the DockQ score and have to resort to model confidence. Figure 2c is an example of a crystal structure being available, demonstrating that AlphaLink indeed improves the prediction quality over AlphaFold-Multimer.

We added new plots that show model confidence \leftrightarrow DockQ score correlation on the challenging SAbDab targets (Extended Data Fig. 10a). Here, model confidence is often overestimated. A second plot shows the relationship between the model confidences of AlphaLink / AlphaFold-Multimer and the resulting DockQ scores (Extended Data Fig. 10b). A better model confidence also results in a better DockQ score AlphaLink over AlphaFold-Multimer.

Most of the targets have only one or two crosslinks (see coloring in Figure 2a), it would

be hard to show a correlation here. The impact of the crosslinking data also depends on the evolutionary data and the position of the links, although in general, with an increase in crosslinks you would see an improvement in prediction quality due to the larger coverage.

Reviewer #3 (Remarks to the Author):

The authors have submitted a natural extension of Alphalink to the modeling of heteromeric complexes. They demonstrate that high quality models can be generated in a fraction of the time of a normal AlphaFold Multimer modeling event. This claim is established with a very nice simulation/sampling of crosslinking data but also backed up in select cases with real (sparse!) data collected from other studies.

In general I find this a very useful and exciting addition, and a very realistic one. Computational methods will likely (always) benefit from the inclusion of real structural data and crosslinking MS is a natural source for such information.

I have only a few matters to raise:

1. As I say, the simulations are a nice way to illustrate the benefits of Alphalink, but some additional information is needed. If I understand correctly, the authors use a span of ~25 angstroms and random multiple samplings of 10 crosslinks, from a set that samples 10% of sequence (at a 20% FDR). I like the challenge presented by these numbers. However, does the sampling of the crosslinks reflect the bias that we often see in real data? First at the level of residue (K much more preferred than STY) and second at the level of structure (links clustered in one area). Additional discussion around the limitations of the simulation would be valuable.

Our simulation could indeed be further improved by reflecting these biases, we currently uniformly subsample the crosslink candidates. This doesn't really affect training because for the model it will not matter if the crosslink originates from K or STY, it's more important to see sufficient data to get enough training signal. And we can see that it translates to the real data. However, it may result in a better coverage in testing than what we can normally expect.

2. The first paragraph after the introduction strikes a rather discordant note. Any claim of "vastly outperforming" should be made after the data are presented, to avoid biasing the reader.

We removed the claim.

3. And I contest the very next statement that claims also that the results are similar or better on CASP15 targets than the best performing algorithms that participated in CASP15. Figure 1a shows a mixed bag, with 50% of the CASP15 algorithms outperforming Alphalink. Please adjust the language on this.

Our new data based on more sampling and recycling leads to an improvement that when contrasted to the CASP15 outcome supports our wording: "Integrating simulated SDA crosslinks in the modelling of eight challenging heteromeric CASP15 targets (H1129, H1134, H1140, H1141, H1142, H1144, H1166, H1167) substantially improved the DockQ15 score from 0.14 to 0.62 on average, compared to the AlphaFold-Multimer baseline (Fig. 1a) which matches the average DockQ = 0.62 of the best predictions in CASP15. For comparison, AFSample16, one of the top performing methods in CASP15, averages a DockQ score of 0.56."

4. And more importantly on this topic, what is to be learned about how and why these algorithms outperform Alphalink? I notice in Extended Data Figure 6 that strongly increasing the sampling/recycling helps only a bit (on one example at least). The authors are perfectly positioned to comment on benefits/weaknesses of approaches.

We re-ran the experiment with parameters that are closer to the other groups in CASP15 to have a fairer comparison. Extended Data Figure 7 (now removed since it is contained in the new Fig. 1a) showed the results only for H1134 for most of the other targets there are larger improvements. We added a discussion of the specific targets where other approaches outperform AlphaLink and possible reasons.

5. Figures:

a. 2C. Is there an actual interaction between RpoA and RpoC in the AlphaFold model? The interface is obscured by the crosslinks. Can this be made more visible?

There is no interaction in the AlphaFold model. We rotated the structures to improve visibility.

b. 3D. I'm a bit puzzled by this success. Can the authors comment how the crosslinking preserved an open state when AlphaFold (which AlphaLink is based on) drives a closed form. Near as I can tell, the orientation of the lower hinge looks the same for both AlphaFold and AlphaLink. Can you prevent an interaction if crosslinks occupy an interface? Would be nice to see some commentary on this and a supporting figure with the crosslinks mapped.

We would like to stress that understanding AI is a very large and active research field that is not close to our current competencies. While AlphaLink bases on AlphaFold, it differs in two ways. It is a different network as it has been re-trained regarding its refinement step (with crosslinks added). It also uses crosslinks from the open conformation.

We left out SamHD because it doesn't have a crystal structure. It's located close to the hinge and will affect the placement since there are also crosslinks. We include SamHD now in the figure.

c. Extended Data figure 1. Better axis labels would be nice.

Fixed!

d. Extended Data figure 6. What does "more" mean exactly? Better labels here too, please. We removed this plot now because the new results in Fig. 1a with increased recycling/sampling supersede it.

Version 1:

Reviewer comments:

Reviewer #1

(Remarks to the Author)

The authors have addressed my comments.

Reviewer #2

(Remarks to the Author)

Reviewer #3

(Remarks to the Author)

The authors have satisfactorily addressed all my concerns. The additional models provided improve the claims that they make.

Author Rebuttal letter:

To say that AFsample doesn't scale is not a very strong argument as the alternative is obtaining crosslinking MS data for a complex of interest which may well be impossible for proteins expressed in low quantities and if substantial overexpression is permitted the proteins may simply interact due to the reason of abundance. Regardless of how long it takes to run AFsample, running a few thousand iterations of predictions will be both faster and cheaper than setting up a lab, buying a MS and expressing the complex of interest.

If ColabDock appeared one month later this is acceptable.

We completely agree that a purely computational strategy holds the advantage over experimentation because its entry barrier is substantially lower. Nonetheless, crosslink studies are being undertaken by an increasing number of labs. Once one has crosslink data, so we argue and show, it is useful to leverage it also for modelling. Crosslinking offers complementary information to evolutionary data, and can thus be leveraged also by an approach like AFsample. Many structural biology labs are willing to invest substantial resources to isolate complexes for cryoEM, all those are very much amenable to crosslinking and then AlphaLink modelling, if size permits. Who knows, maybe even top computational labs like the Wallner lab might find it attractive to align with an experimental lab to invest into acquiring crosslink data should they find interest in specific biological challenges.

To say that there is no overlap between the different datasets due to them being from different dates shows how little the authors understand about the concept of benchmarking which is

evident throughout the presented work. It is not only important to create a useful method but also to demonstrate its performance in a meaningful way. Date cutoffs show no information regarding the overlap between training and test sets. What is important is to be able to predict new complexes, not reiterate already known ones. No proof for the claim that the potential overlaps are negligible are presented. To ensure that the method is meaningful you absolutely need to display the performance on unseen targets (no homology <20% seqid or better TMscore<0.5).

As the reviewer points out, there are different levels of overlap (e.g., targets being part of the training set, homologous targets in the training set, using templates in prediction). We clarified in the text what we are talking about and what possible overlaps are present. Importantly, any overlap improves the AF baseline performance and thus works against us by reducing the possible gains from crosslinks. The goal is to show that integrating crosslinking MS data into AlphaFold-Multimer can improve the prediction quality on challenging targets. In our direct comparison with AlphaFold-Multimer, both methods use the same input data (minus or plus crosslinking MS data). The AlphaLink training set includes some newer targets in comparison to AlphaFold-Multimer v2.2. We showed that the additional training (fine-tuning) of AlphaLink wasn't the reason for the increased performance (Extended Data Figure 7) but it's the addition of crosslinking MS data which substantially improves the prediction quality on many targets over AlphaFold-Multimer (Fig. 1).

The CASP15 targets we benchmark on include unseen targets since they consist of TBM/FM targets, meaning there are only partial templates available for a subunit and / or the assembly. Antibody-antigen targets are challenging in their own right due to the lower evolutionary signal. Running a job for 16h is not very long. How long does it take to get new cross links for a complex of interest? This is the question to be asked to answer how useful the proposed method is.

A couple of days. It's 16h for a single prediction. So in this example, 1000 samples will take almost 2 years to compute on a single A100 GPU. The runtime scales cubically.

We would disagree. This is not the question to ask to determine the "usefulness". It is certainly one aspect though and will help with larger complexes where sampling becomes much more expensive. Crosslinking MS data provide additional information and add some degree of experimental validation to the prediction. They are complementary to evolutionary information which for many interesting targets are lacking/ sparse (e.g., antibody-antigen complexes). Fig 1b) demonstrates the usefulness for these targets.

While the primary goal is not to convert computational titans, such as Wallner's group, into crosslink-focused facilities, it is important to demonstrate the significant advantages of utilizing crosslink data in structural biology. As more labs acquire crosslink data, AlphaLink can effectively harness this information to enhance the modelling of complexes. Our work demonstrates that integrating crosslink data, already increasingly common in structural biology, can lead to more accurate and insightful predictions in AlphaLink.

Perhaps you can expand a little on what XWalk does (I am not familiar with the package). As I understand it, it uses structural factors to infer crosslinks. What structures do you input to the package and what information from these structures is used to "simulate" the crosslinks? If I interpret correctly, you input the native complexes that you later use for evaluation(?). There is therefore obvious bias in this evaluation. You can't input information from the native complex structures and then claim that you predict them and I see little utility in being able to do so. We use XWalk to computationally generate crosslink data for the training and evaluation complexes. Generating experimental data for all these complexes is currently unfeasible. Please realise that we need to get crosslink data somehow for our approach. XWalk identifies residue pairs in the native complex that could be crosslinkable (here with SDA, < 25 Å) based on solvent accessibility and peptide digestion. We select at random a fraction of these pairs and add noise in line with real experiments. This computational crosslink data resembles closely that of real experiments.

Again, cutoff dates are not meaningful here. If you want to prove that your method actually works, you can't do so on structures homologous to those seen by AFM - regardless of when they were released. Please partition the data properly and then do an evaluation. As it is now, it is not possible to assess the utility of your method on new targets which I presumed was the idea.

We find cutoff dates meaningful here because they ensure that we did not directly train on targets from CASP15 or included templates of these targets (same with SAbDab). The evaluation setup is therefore exactly the same as it was in CASP15 which helps to assess the utility of our method, namely improving prediction quality on challenging targets with crosslinking MS data. The CASP15 benchmark targets consist of TBM/ FM targets, meaning only partial templates are available. Repartitioning the data only works if you actually train from scratch, which is both impractical and unfeasible (resources) and would disqualify AlphaFold-Multimer as a comparison point. We would like to stress again that we wanted to demonstrate the additional value offered by adding crosslink information and therefore conduct a comparison. Any advantage for AlphaFold baseline is simply reducing the impact of crosslinks. So, any bias will

work against us and not for us. The positive effects we observe are despite any biases from structure overlap.

This can (and should) be easily checked.

AlphaFold was trained on all targets before the cutoff and may include some homologs after the cutoff date.

Again with the date cutoffs. It seems that you really do not understand the importance of partitioning data for ML. If the data is similar between the two dates, it is impossible to say how well your method generalises.

Please see above.

Redo the data partitioning. This is something you have to consider before training. I can't accept the release of another paper that simply ignores overlaps between train and test sets and instead relies on what day in the week it is (date splits). Too many papers in this space have been published without considering these issues which you and others (editors among them) seem completely unaware of.

Redoing the data partitioning only works if you actually train from scratch which we have not done. As said above, we compare AlphaFold-Multimer with AlphaLink2 with the weights of AlphaFold-Multimer as a starting point for fine-tuning, so essentially AF +/- crosslinks.

How can you claim that the correlation is good when it is not? It is evident that the confidence is high regardless of DockQ score:

[Redacted]

If you plot confidence vs DockQ as I suggest you will see this more clearly. The model confidence is high all over the place and even at DockQ scores as low as 0.1 the confidence is 0.7-0.8.

We stated that it was shown in the AlphaFold-Multimer paper that the ipTM (> 0.6) correlates well with the DockQ score, which in turn is close to the model confidence. This plot actually shows the ranking performance. We fixed the title and removed the colouring by model confidence. The confidence vs DockQ is plotted in Extended Data Figure 10a.

Great that you here finally admit to this. This should be in the main text as a strong limitation. Currently, there is no way to separate accurate from inaccurate predictions making the utility of the predicted models questionable. A reason why methods like AFM and AFsample are useful is that the confidence can be used to select accurate models from many predictions.

Extended Data Figure 6 shows that the model confidence can be used to select accurate models from many predictions. Most of the time, the top ranked prediction is very close to the best prediction, and this gets better with higher model confidence (similar to AlphaFold-Multimer, ipTM > 0.6).

Our direct comparison with AlphaFold-Multimer shows that a higher model confidence for AlphaLink seems to indicate higher model quality (Extended Data Fig. 10b).

Incidentally, AFsample seems to be a poor example. The two plots in the AFsample paper (Fig 1. c,e) show overall a fairly bad correlation between DockQ and ranking_confidence.

If this is the case, please provide support for this statement by calculating and visualising the score differences divided by the number of cross links. You could e.g. plot the score distributions having 1,2,... crosslinks.

Our claim is that adding available crosslinks to AlphaFold-Multimer is generally beneficial. This claim is substantiated by our presented data. That more information leads to better models has been proven in many instances. We have not extended this to crosslinks in our manuscript, however, because of the unfavourable cost/benefit ratio.
